# Healing Evaluation of Asphalt Mixtures with Polymer Capsules Containing Rejuvenator under Different Water Solutions

**Zhifeng Li** [1], **Huan Wang** [1], **Pei Wan** [1,*], **Quantao Liu** [1,2], **Shi Xu** [3,4], **Jian Jiang** [5], **Lulu Fan** [5] and **Liangliang Tu** [5]

1 State Key Laboratory of Silicate Materials for Architectures, Wuhan University of Technology, Mafangshan Campus, Wuhan 430070, China; 321066@whut.edu.cn (Z.L.); 303561@whut.edu.cn (H.W.); liuqt@whut.edu.cn (Q.L.)
2 Wuhan University of Technology Advanced Engineering Technology Research Institute of Zhongshan City, Zhongshan 528400, China
3 School of Civil Engineering and Architecture, Wuhan University of Technology, Mafangshan Campus, Wuhan 430070, China; xushi@whut.edu.cn
4 Faculty of Civil Engineering and Geosciences, Delft University of Technology, Stevinweg 1, 2628 CN Delft, The Netherlands
5 Shenzhen Sez Construction Group Co., Ltd., Shenzhen 518034, China; jiangjian@szcg.cn (J.J.); fanlulu@szcg.cn (L.F.); tlliang@whut.edu.cn (L.T.)
* Correspondence: wanpei@whut.edu.cn

**Abstract:** Polymer Ca-alginate capsules with rejuvenator bring a high healing level for asphalt concrete under dry healing environments; however, the healing levels of bituminous mixtures containing capsules under water healing conditions are still unknown. In view of this, this study aimed at exploring the healing levels of asphalt concrete containing polymer capsules under various solution healing conditions following cyclic loads. This study involved the preparation of capsules, followed by the evaluation of their morphological characteristics, resilience to compression, thermal endurance, and rejuvenator content. The assessment of the healing properties of asphalt concrete utilizing capsules was conducted through a fracture–heal–refracture examination. This study conducted Fourier transform infrared spectrum experiments to determine the rejuvenator release ratio of capsules under dry conditions and the remaining rejuvenator content in extracted bituminous binder from capsule–asphalt concrete after solution treatment. Meanwhile, a dynamic shear rheometer was utilized to investigate the rheological characteristics of asphalt binder. Results revealed that the healing ratios of capsule–asphalt concrete beams under a dry healing environment were significantly higher than that of beams under various solution healing conditions, and the alkali solution has the worst effect on the improvement in healing ratio. The coupled impact of moisture intrusion and ion erosion resulted in an enhancement of complex modulus of asphalt binder while concurrently reducing its phase angle. Consequently, the restorative capacity of the asphalt binder was weakened.

**Keywords:** polymer capsules; asphalt concrete; healing ratio; solution environment





## 1. Introduction

Due to superior performance in terms of traffic safety, driving stability, noise management, and moisture resistance, asphalt mixes are frequently employed in the building of motorways and urban roads [1,2]. However, asphalt pavement inevitably experiences microcracks and ageing owing to various unfavorable conditions such as abrupt temperature changes [3], ongoing traffic volume [4], unremitting air oxidation [5], water erosion [6], and intense UV exposure [7]. Inadequate and untimely maintenance of asphalt pavement can result in the exacerbation of microdamage, leading to the development of macrodamage. This can significantly curtail the service life of the concrete framework and compromise the structural stability of the building foundation, thereby posing a potential threat to the safety of vehicular traffic. At present, traditional maintenance methods are classified

as passive repair approaches that are implemented after the occurrence of macroscopic road destruction [8]. These methods are associated with significant drawbacks, including prolonged maintenance construction periods, substantial expenses for maintenance, brief life span, as well as a certain degree of environmental pollution [9–12]. Consequently, there is an urgent requirement for ecologically friendly premaintenance innovations in the context of asphalt pavement.

Bitumen, a viscoelastic material, has been found to exhibit temperature sensitivity. It can repair its inside microcracks through a process of self-healing when exposed to adequate rest periods or elevated temperature conditions [13–16]. Nonetheless, this desired healing state is not typically achievable in regular usage environments owing to unstoppable traffic flow and uncontrollable meteorological conditions, which presents practical challenges for the autonomous curing of internal cracks within asphalt materials. To improve the healing ability of asphalt, some researchers have tried to add polymer, biomaterials, and nanoparticles into asphalt [17,18]. However, the obtained healing ability was low. Therefore, the development of effective maintenance techniques that leverage the improved self-healing capabilities of asphalt materials is of utmost importance.

In contemporary times, several distinctive remedial methodologies have been formulated to expedite the recuperative mechanism of asphalt substances, primarily encompassing thermal induction healing [19–22] and rejuvenator-induced healing [23–28]. The thermal healing technique involves two main methods: electromagnetic induction heating and microwave-induced heating. These methods operate by externally exciting conductive or microwave-absorbing substances into bituminous mixtures, resulting in an increase in asphalt body temperature following external activation [29]. This elevated temperature facilitates the movement of asphalt surrounding the microcrack region, ultimately leading to the closure of cracks in asphalt concrete.

In contrast to conventional maintenance approaches, the utilization of thermally induced self-healing innovation has demonstrated a notable capacity to enhance the repair of microcracks in asphalt pavement. However, certain limitations have been identified, which impede its extensive implementation. For one, this technique necessitates an enormous amount of outside energy to function properly, which not only utilizes a large mass of renewable resources but also generates greenhouse gases, consequently making it incompatible with the concept of ecological pavement upkeep. Furthermore, this approach, centered on augmenting healing temperature, fails to effectively tackle the asphalt ageing issue and may expedite the ageing phenomenon. Rejuvenator encapsulation technological advance is considered a viable approach for expediting the repair of microcracks in asphalt and selectively revitalizing the deteriorated asphalt surrounding the cracks. This technology employs healing agent that comprises abundant light asphalt components, which facilitate the restoration of the unique natural healing features of the aged asphalt binder [30]. The rejuvenator encapsulation innovation is a self-healing mechanism that has been improved through implementation of the healing agent's smart release.

Encapsulating diverse asphalt-regenerating agents within capsules is a prevalent method for enhancing the healing ability of bitumen. The predominant storage modalities for asphalt rejuvenator consist of nucleus–shell microcapsules featuring micrometer dimensions and multichamber capsules with millimeter dimensions. The incorporation of microcapsules containing rejuvenator exhibited significant enhancement in the healing capacity of asphalt binder, as observed using the fracture recovery test [31–33]. The efficacy of crack repair in asphalt concrete utilizing microcapsules is limited due to an insufficient rejuvenator amount and temporary release manner. A previous study found that usage of multichamber calcium alginate capsules demonstrated an obvious ability to gradually release rejuvenator, resulting in sustained healing capacity for asphalt concrete when subjected to outside cyclical stress [26,27,34,35]. Therefore, the integration of rejuvenator-containing calcium alginate capsules into bituminous concrete represents a promising approach for achieving long-lasting asphalt pavement.

The ongoing research on asphalt mixtures that incorporate calcium alginate capsules for self-healing purposes are noteworthy due to their execution under controlled non-aqueous healing conditions [15,36]. The impact of moisture conditions on the crack repair of asphalt concrete with various rejuvenators was investigated by Riara et al. [37]. The study finding indicated that the influence of water environment on the healing of asphalt cracks within the temperature range of 25–45 °C was negligible when compared to the dry environment.

However, the performance degradation of asphalt and asphalt mixture can be accelerated by variations in service surroundings, for instance, areas with high precipitation, sodium hydroxide-alkali soil, and acidic rainfall, which can alter the water composition environment. Nonetheless, there is a dearth of scientific inquiry regarding the effects of different water compositions present in various environments on the self-restorative properties of asphalt mixtures that incorporate enclosed healing agents. Hence, it is vital to study the effect of different water environments on the crack repair performance of asphalt concrete containing calcium alginate capsules. In view of this, this paper aimed at investigating the healing properties of asphalt concrete with Ca-alginate capsules under different water solution conditions after cyclic loading. The polymer capsules were first produced through the employment of the hole coagulation bath method. Following this, performance assessments were carried out to assess the basic properties of the prepared capsules. This study conducted an evaluation of the strength recovery rates of asphalt concrete with capsules under varying water compositions, including distilled water, acid, salt, and alkali solution. This was achieved through the implementation of the three-point bending test and cyclic load healing test. The healing agent release levels of capsules embedded in asphalt concrete were determined using Fourier transform infrared spectroscopy. The present study investigated the rheological characteristics of asphalt binder following exposure to varying water compositions, utilizing a dynamic shear rheometer. The research methodology of this paper is shown in Figure 1. This research provides preliminary experimental support for the healing performance of asphalt concrete with capsules in an aqueous service environment. Meanwhile, it also provides technical support for the engineering application of capsules in the future.

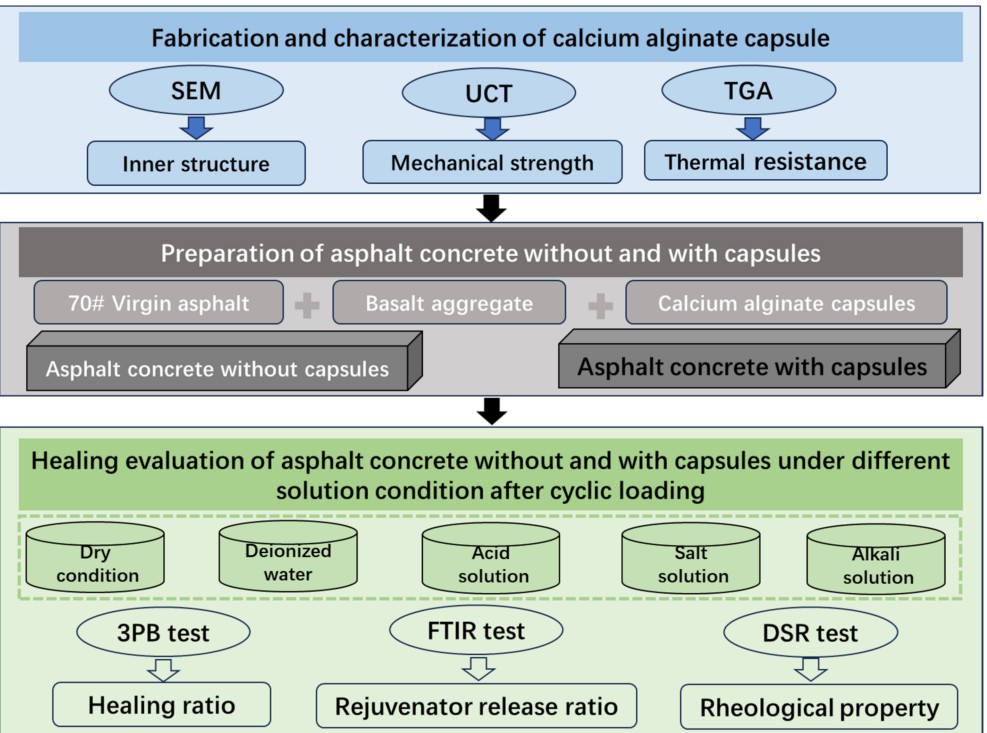

**Figure 1.** The research methodology of this research.

## 2. Materials and Experimental Methods

### 2.1. Raw Materials

Polymer capsules were made from sodium alginate, calcium chloride, Tween 80, and sunflower oil as basic ingredients. Meanwhile, the raw materials used for preparing various solutions were $H_2SO_4$, NaCl, and NaOH. Table 1 displays the fundamental material details. This study reports the values of density, viscosity, penetration, and softening point for virgin asphalt (#70) as 1.034 g/cm$^3$, 231.3 Pa·s (60 °C), 68 (0.1 mm, 25 °C), and 48.4 °C, respectively. The basic information of asphalt rejuvenator sunflower oil is presented in Table 2. Sunflower oil can assist in reestablishing the natural healing properties of aged asphalt by providing a light component [38–40]. Furthermore, sunflower oil exhibits a definite peak at 1745 cm$^{-1}$, whereas there is an absence of any absorption peak within the range of 1700 and 1800 cm$^{-1}$ [27,28,34,41]. Consequently, FTIR spectra reveal that this unique peak (1745 cm$^{-1}$) may be utilized to measure the rejuvenator discharge ratio of capsules within asphalt concrete following repeated loading cycles.

**Table 1.** The fundamental information of raw materials used in this work.

| Raw Material | Purity | Supplier |
|---|---|---|
| Sodium alginate | CP | Sinopharm Chemical Reagent (Shanghai, China) |
| Anhydrous calcium chloride | CP | Sinopharm Chemical Reagent (Shanghai, China) |
| Tween 80 | AR | Sinopharm Chemical Reagent (Shanghai, China) |
| Sunflower oil | Food grade | Arowana Group Co., Ltd. (Beijing, China) |
| $H_2SO_4$ | CP | Sinopharm Chemical Reagent (Shanghai, China) |
| NaCl | CP | Sinopharm Chemical Reagent (Shanghai, China) |
| NaOH | CP | Sinopharm Chemical Reagent (Shanghai, China) |

**Table 2.** The basic properties of sunflower oil used in this work [42].

| Item | Value |
|---|---|
| Appearance | Light yellow liquid |
| Main component | Fatty acid |
| Density (15 °C) | 0.935 g/cm$^3$ |
| Viscosity (60 °C) | 0.285 Pa·s |
| Flash point | 230 °C |

### 2.2. Preparation of Polymer Capsules

The polymer capsules were fabricated under normal circumstances, and the precise methodology is illustrated in Figure 2. The procedure was segmented into four distinct stages: (1) A 2.25 wt% even sodium alginate (SA) solution of was prepared by adding sodium alginate powder into water at ambient temperature and stirring for 5 min. (2) Tween 80 and rejuvenator were incorporated into the prepared sodium alginate solution, followed by subjecting the mixture to shear at 5000 rpm for a duration of 15 min, resulting in the formation of a uniform emulsion of sodium alginate and oil (SA-O). The experimental conditions involved a constant ratio of water to oil at 1:10 and a surfactant amount of 5% based on the mass of oil. (3) The finalized emulsion was transferred into a bespoke funnel and launched into $CaCl_2$ solution (3.0 wt%). The solution was then allowed to rest for a duration of 12 h to guarantee a full reaction between the alginate chains and $Ca^{2+}$. (4) The damp capsules were extracted, subjected to a water rinse, and subsequently placed in a dish at an ambient temperature for a duration of 48 h to achieve a state of dryness.

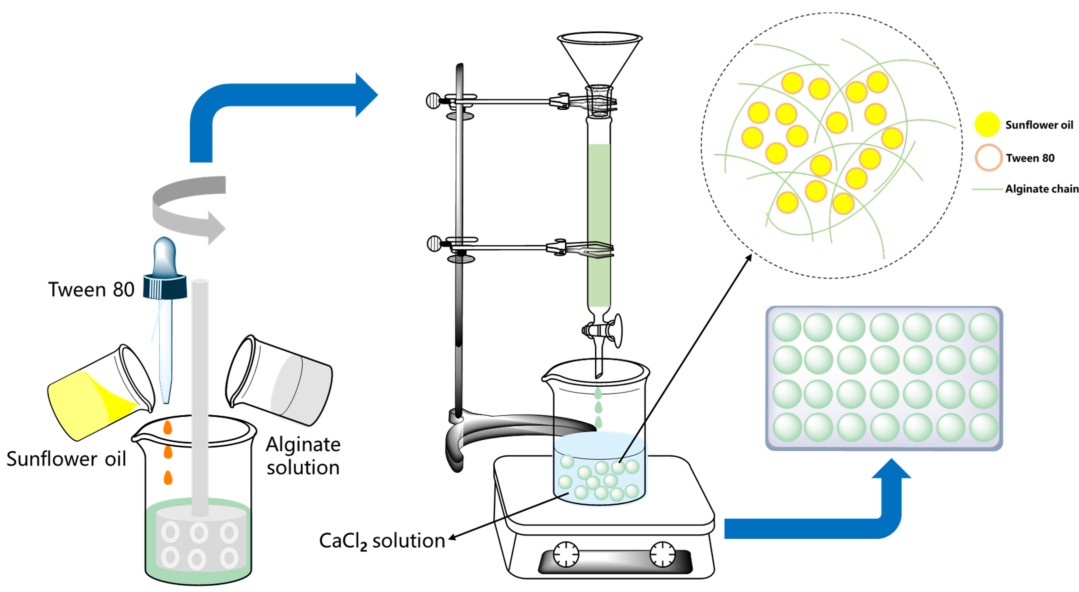

**Figure 2.** The preparation instruction of polymer capsules.

### 2.3. Performance Evaluation of Polymer Capsules

The prepared capsules experienced characterization experiments to ascertain their fundamental performance. The microstructure within the capsules was analyzed using scanning electron microscopy. The uniaxial compression test was utilized to evaluate the compression strength of calcium alginate capsules. Since temperature has an impact on the mechanical strength, prior to the test, three distinct categories of capsules were subjected to storage conditions of 25 °C for 4 h and 160 °C for 2 h. The temperature response property of the capsules was determined using a simultaneous thermal analyzer. The determination of the rejuvenator content of capsules was achieved by computing the residual masses of the empty capsules and the rejuvenator. Table 3 presents the fundamental details of the testing apparatus.

**Table 3.** The basic information of the instruments used in this work.

| Test | Origin | Test Parameter |
|---|---|---|
| SEM | Gemini 300<br>Zeiss<br>Oberkochen, Germany | Shooting area: Capsule section<br>Spray material: Pt powder<br>Spray time: 30 s |
| UCT | Instron 5967<br>Instron<br>Norwood, America | Loading speed: 0.5 mm/min |
| TGA | STA449F3<br>Netzsch<br>Selb, Germany | Temperature: 40–800 °C<br>Heating rate: 10 °C/min<br>Protect gas: $N_2$<br>Injection rate: 20 mL/min |
| FTIR | Nicolet 6700<br>Thermo Fisher Scientific<br>Waltham, MA, America | Chip type: KBr<br>Scan range: 4000–400 cm$^{-1}$<br>Scan time: 64 cycles |
| DSR | Smartpave 102<br>Anton paar<br>Graz, Austria | Strain: 0.5%<br>Frequency: 10 rad/s<br>Temperature: 30–80 °C<br>Plate diameter: 25 mm |

### 2.4. Fabrication of Asphalt Concrete with Polymer Capsules

The present study opted for a compact asphalt mixture, and the corresponding gradation (AC-13) is illustrated in Figure 3. At the conclusion of the mixing process, Ca-alginate capsules were incorporated into the asphalt concrete, with a weight equivalent to 0.5% of

the overall weight of the entire asphalt mixture. Following the mixing protocol, conventional rutting-resistant asphalt concrete plates were produced alongside capsule-containing counterparts using a specialized slab-forming apparatus. Subsequently, beams of bituminous mix measuring 98 mm × 45 mm × 50 mm were extracted from the asphalt concrete blocks. A notch measuring 5 mm × 4 mm was then created in the central region.

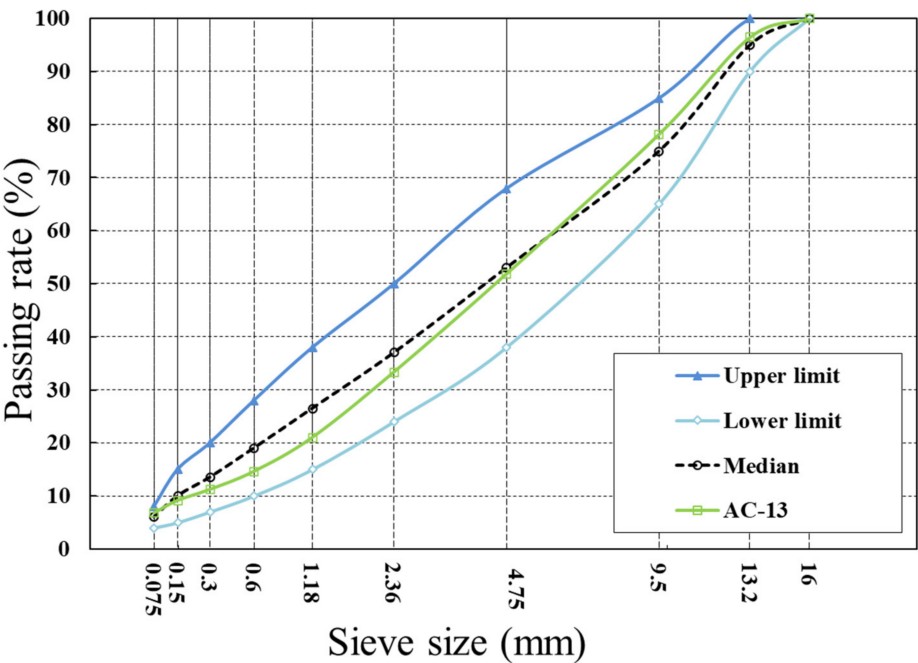

**Figure 3.** The aggregate gradation curve of AC-13 bituminous mixtures [42–44].

### 2.5. Healing Efficacy Measurement of Asphalt Concrete with Polymer Capsules

The experimental procedure for the fracture healing test was partitioned into four distinct stages, as depicted in Figure 4. (a) In order to determine the original bending strength, a series of 3-point bending (3PB) tests were conducted in a sequential manner on asphalt mixture beams, both with and without capsules. The experimental conditions involved setting the environmental temperature to −20 °C and the compression load application rate to 0.5 mm/min. (b) The fractured beam was inserted into a steel casting and subjected to an even dispersion of pressure (0.7 MPa) through the placement of metallic plates on the beam. The load periods were configured at intervals of 0, 16,000, 32,000, and 64,000 cycles. The experimental conditions involved setting the test temperature to 20 °C. The capsules were induced to discharge the enclosed rejuvenator through the application of an external load. (c) Following the conclusion of cyclic loads, the beams were extracted from the steel molds and immersed in distinct solutions, namely deionized water, $H_2SO_4$ @pH = 3, NaCl @pH = 7, and NaOH @pH = 11. Subsequently, the beams were allowed to rest for varying periods of time, specifically 48 h, 72 h, and 96 h, at a temperature of 20 °C, with the aim of recuperating their strength. (d) Upon completion of the recovery phase, the healed beams were subjected once more to the 3PB examination, as outlined in step 1.

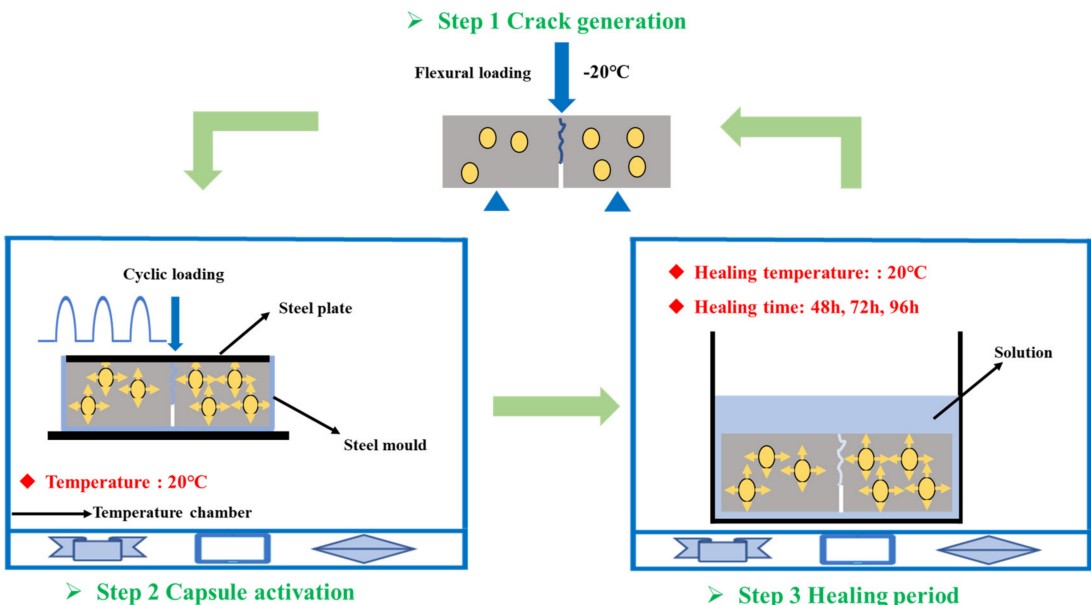

**Figure 4.** Healing assessment test procedure of asphalt concrete used in this work.

The healing degree of cured asphalt concrete was assessed through the strength restoring percentage ($HI_S$). The definition of the $HI_S$ was established by determining a particular value between the first bending strength ($F_1$) and the beam's bending strength following the rest duration ($F_2$), as indicated in Equation (1). This study involved the testing of three the samples in every group to determine the mean degree of healing.

$$HI_S = F_2 / F_1 \tag{1}$$

*2.6. Quantification of Loading-Released Rejuvenator Content in Asphalt Binder Release from Capsules after Different Solution Treatments*

The present study utilized an FTIR instrument to conduct a chemical analysis of the therapeutic agents discharged using capsules in asphalt mixtures. The infrared spectrum of sunflower oil exhibits a prominent peak at 1745 cm$^{-1}$, which corresponds to the amount of oil content within bitumen. Conversely, asphalt does not exhibit any peak within this range. Asphalt specimens were made by blending oil along with asphalt at a temperature of 120 °C for a duration of 40 min. The rejuvenator was added in varying proportions of 0%, 2%, 4%, 6%, and 8% of the asphalt weight. The asphalt samples that were prepared underwent FTIR testing to establish the relationship among the area of 1745 cm$^{-1}$ and the percentage of sunflower oil in the asphalt binder. The study conducted by Rao et al. revealed that the utilization of the peak index outlined in Equation (2) is a viable approach for ascertaining the quantity of rejuvenator present in the asphalt binder [26]. Figure 5 presented the correlation curve between $I_{1745cm^{-1}}$ and the oil content of asphalt. It can be observed that the index $I_{1745cm^{-1}}$ raised the healing agent content within the asphalt binder and presented a linear relationship ($R^2$ = 0.9780).

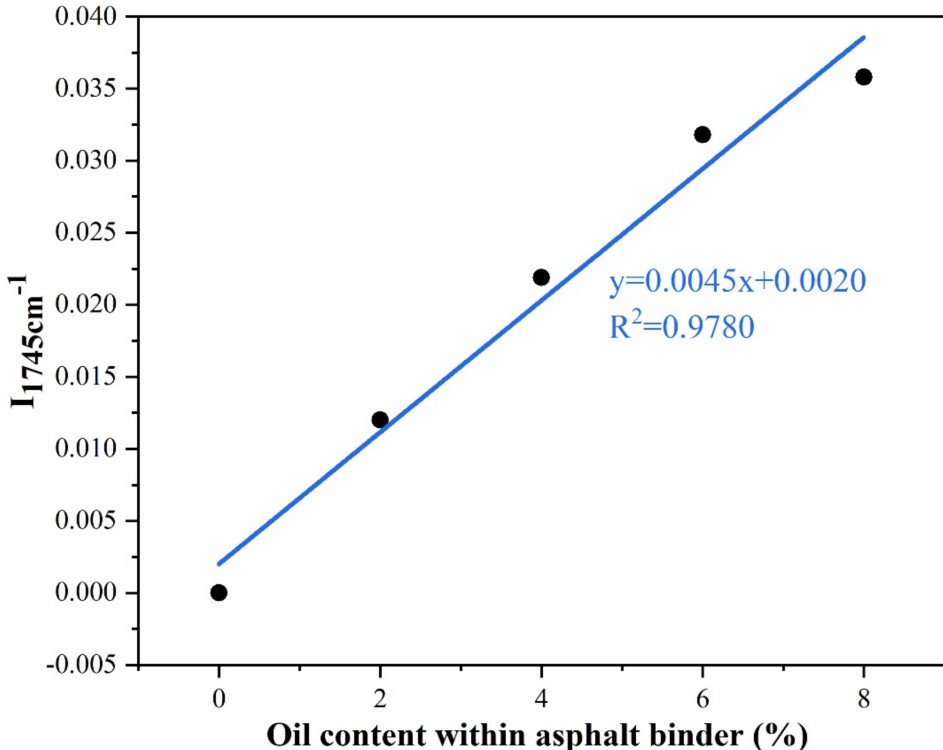

**Figure 5.** The standard curve between $I_{1745cm^{-1}}$ and rejuvenator content within asphalt [26].

The experimental beams underwent a repeated loading–healing procedure, after which they were subjected to an 80 °C heating process for a duration of 45 min. The capsules present within the asphalt mix were subsequently extracted manually. The asphalt mixtures that were not tightly bound were subjected to dissolution in trichloroethylene for a duration of 48 h. The resulting supernatant was collected and subsequently placed in a fuming cabinetry for 24 h to facilitate the evaporation of the solvent. A 0.1 g quantity of asphalt was introduced into a centrifuge tube, followed by the addition of 2 mL of $CS_2$ to disperse the bitumen. Asphalt oil was applied onto KBr wafers and subsequently dehydrated to generate an asphalt layer. The FTIR experiments were performed in the mid-infrared range of wave numbers, specifically between 400 and 4000 $cm^{-1}$. The parameters for the depth of field and overall check duration were established as 4 $cm^{-1}$ and 64, respectively.

$$I_{1745cm^{-1}} = \frac{\text{The peak area of 1745 cm}^{-1}}{\sum \text{Area of spectral bands between 2000 and 600 cm}^{-1}} \tag{2}$$

### 2.7. Rheological Characteristics Test of the Asphalt Binders Extracted from Asphalt Concrete

The rheological properties of the retrieved asphalt binders were characterized through temperature scan tests using a dynamic shear rheometer (DSR). Test samples were selected as the extracted asphalt binder after undergoing 64,000 cycles of compressive loading using different solution treatments. The operating temperature range utilized in the DSR test ranged from 30 to 80 °C. The magnitude of the strain was measured as 0.5% and the angular frequency was determined as 10 radians per second. The rotor's diameter measured 8 mm.

## 3. Results and Discussion

### 3.1. Fundamental Characteristics of Prepared Polymer Capsules

3.1.1. Morphological Structure

Figure 6 depicts the exterior appearance and inner construction of polymer capsules. As depicted in Figure 6a, the capsules exhibited nearly spherical morphology. Moreover, it is evident from the graphical representation in Figure 6b–d that the capsules exhibited a complex multicavity inner structure. The therapeutic droplets (sunflower oil) were preserved within discrete cavities of varying sizes and shapes. The distinct preservation mechanism of the capsule has the potential to gradually dispense its internal therapeutic agent in response to external pressure. The present study conducted a statistical size analysis of 50 individual polymer capsules (as shown in Figure 7) and revealed that the mean diameter of capsules containing rejuvenator was 1.66 mm. Therefore, the capsules have the potential to be incorporated into asphalt mixtures as a partial replacement for fine aggregates.

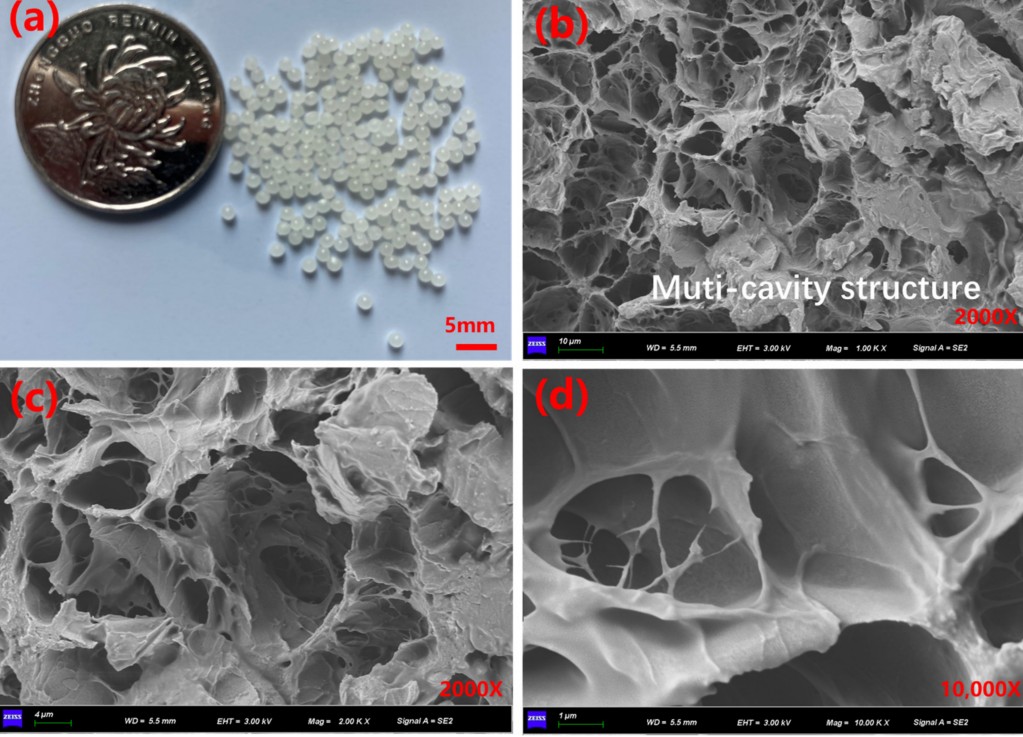

**Figure 6.** (**a**) Appearance image of polymer capsules; (**b–d**) inner structure of capsules at varying magnification levels.

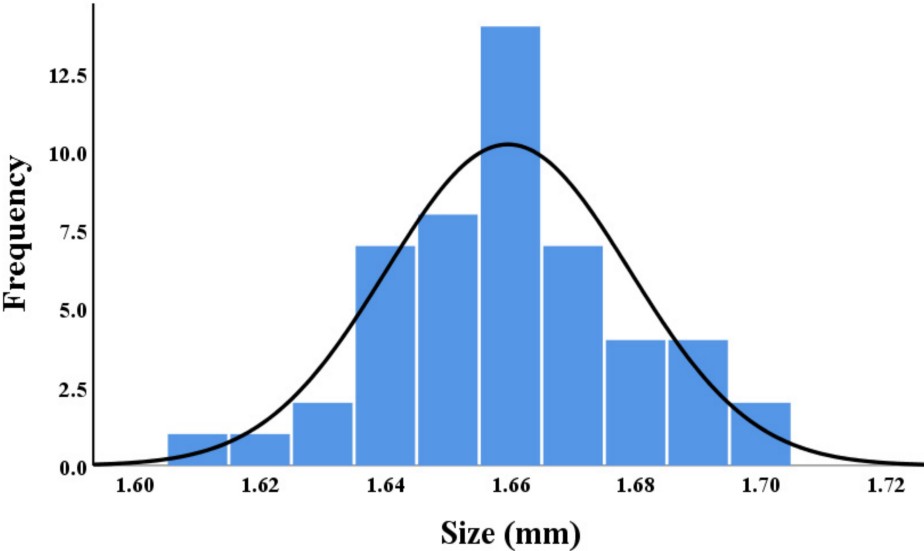

**Figure 7.** The size distribution of polymer capsules used in this work.

### 3.1.2. Mechanical Strength

The capsules, with millimeter size, are intended to be incorporated into bituminous mixtures as constituent components of the fine aggregates. Therefore, it is imperative that the polymer capsules exhibit mechanical durability during the production procedure of asphalt concrete. Empirical evidence indicates that the capsules utilized in bituminous mixtures necessitate a mechanical strength exceeding 10 N. Figure 8 illustrates the yield strength of the capsules after undergoing a particular temperature intervention. This study investigated the impact of various temperature treatments (25 °C and 160 °C) on the mechanical strength of the capsules. Our findings imply that the capsules' mechanical strength decreased as the temperature increased, with values of 12.8 N and 10.6 N recorded for the respective treatments. The strength of the capsules declined slightly due to the softening caused by an increase in temperature. The capsules exhibited a strength exceeding 10 N even when subjected to temperatures under 160 °C, indicating their viability for use in asphalt concrete production.

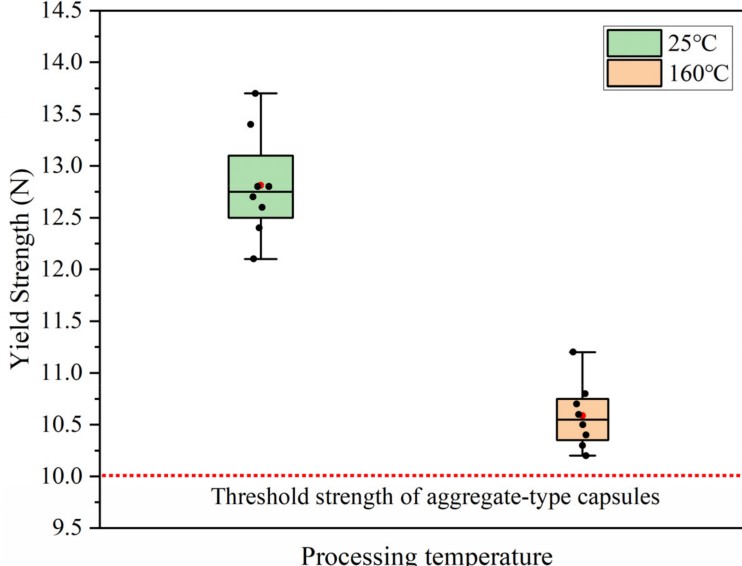

**Figure 8.** The compression strength of the polymer capsules.

### 3.1.3. Thermal Stability and Relative Rejuvenator Content

The incorporation of polymer capsules into bituminous mixtures occurs after the mixing period, followed by ongoing compression. The capsules must exhibit the ability to endure the elevated temperatures generated during the production procedure. It is imperative that the temperature at which calcium alginate capsules undergo thermodynamic disintegration surpasses the temperature at which asphalt mixtures are produced. Thermal gravimetric analysis (TGA) is a widely employed technique for examining the thermal durability of capsules through the determination of their mass reduction at elevated temperatures.

Figure 9 displays the weight declines of the sunflower oil, empty polymer capsules (lacking healing agent), and polymer capsules with healing agent. According to the data presented in Figure 9a, the process of sunflower oil volatilization commences at a temperature of 345 °C and reaches completion at 498 °C. Therefore, the residual weight of both empty capsules and oil capsules within the temperature range of 345 to 498 °C can serve to ascertain the healing agent content of the capsules. Figure 9b illustrates the remaining mass of empty polymer capsules, along with their residual weight at temperatures of 345 °C and 498 °C, which were found to be 50.1% and 40.9%, respectively. Figure 9c illustrates the remaining weight that was retained by the polymer capsules containing oil. The residual mass of the capsules exhibited a stable decrease within the temperature range of between 100 °C and 180 °C. This occurrence can be ascribed to the process of moisture vaporization and the degradation of minute glycosidic bonds present in the alginate chains. The findings indicate that the capsules containing rejuvenator exhibited a mass loss of 4.3% when subjected to an ambient temperature of 200 °C, which surpasses the temperature utilized in the production of bituminous mixtures. Overall, the polymer capsules exhibited advantageous thermal endurance when subjected to the manufacturing temperature of asphalt mixtures. Additionally, the residual masses of the capsules at temperatures of 345 °C and 498 °C were found to be 86.3% and 17.2%, respectively. Hence, the computation of the quantity of the active ingredient presented in the capsules can be ascertained by means of Formulas (3) and (4). The percentage of relative rejuvenator content found in the capsules was 58.6%.

$$\lambda + (\delta_1 - \delta_2)(1 - \lambda) = \eta_1 - \eta_2 \tag{3}$$

$$\lambda = \frac{\eta_1 - \eta_2 + \delta_2 - \delta_1}{1 + \delta_2 - \delta_1} \tag{4}$$

where $\lambda$ is the relative oil content in the capsule (%), $\delta_1$ and $\delta_2$ are the remaining weight of empty capsules at 345 °C and 498 °C, respectively, $\eta_1$ and $\eta_2$ are the retained weight of the capsules with oil at 345 °C and 498 °C, respectively.

### 3.2. Healing Evaluation of Asphalt Concrete

### 3.2.1. Healing Ratio of Asphalt Concrete following Various Solution Treatments

The healing ratios of bituminous mixture beams without and with capsules following cyclic loads and different water solution treatments are shown in Figures 10 and 11, respectively. The healing temperature and healing time were controlled at 20 °C and 48 h. As can be observed from Figure 10, the whole noncapsule beams achieved partial strength recovery owing to the inherent healing character of asphalt binder. Meanwhile, the inherent capacity of noncapsule beams to recuperate their strength was observed to diminish upon the implementation of water solution treatment. The reason is that the intrusion of moisture leads to a reduction in cohesive force between asphalt layers and the adhesion stress between asphalt and aggregate [6,45,46]. The salt (NaCl) and alkali solution (NaOH) have an obvious adverse effect on the natural healing ability of asphalt concrete. The intrusion of sodium salt solution into the asphalt and subsequent crystallization results in the obstruction of asphalt flow [47,48].

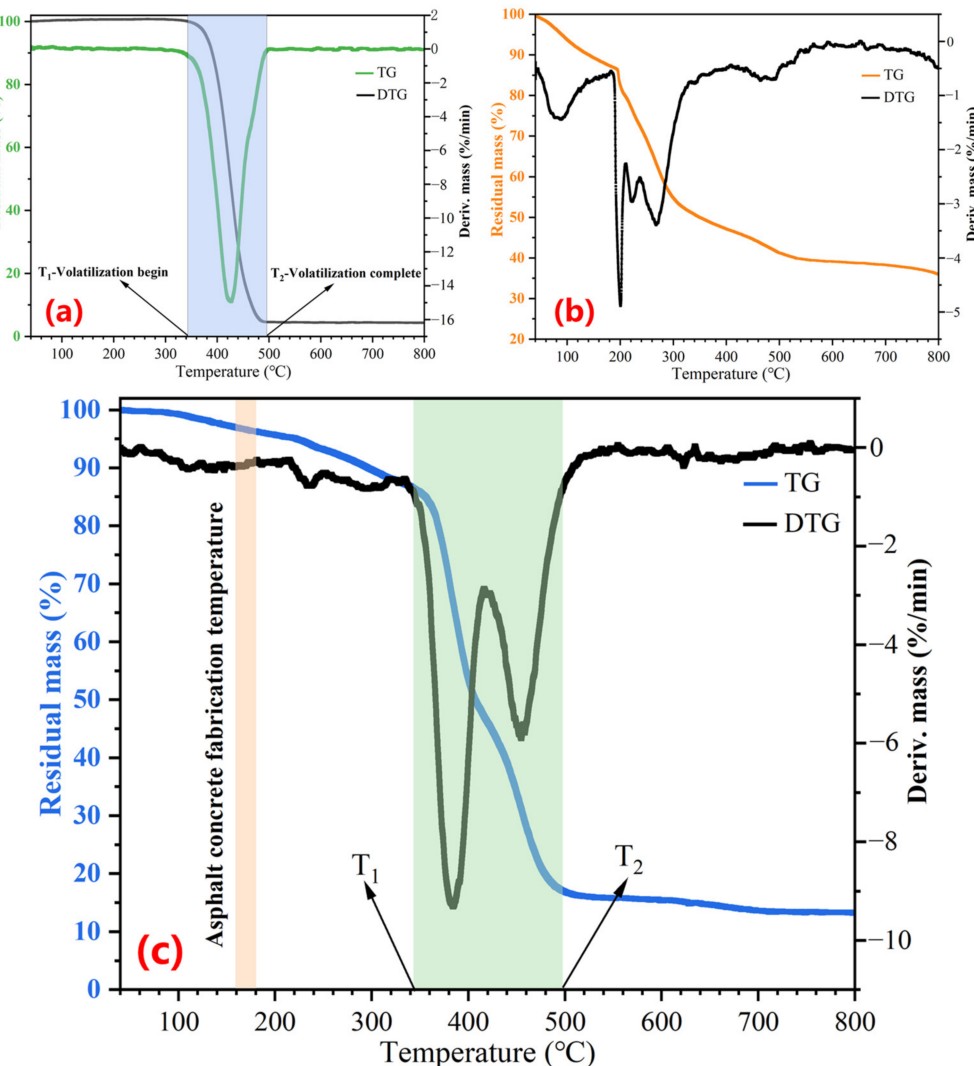

**Figure 9.** The remaining weight of various materials: (**a**) sunflower oil, (**b**) empty polymer capsule, and (**c**) capsule containing oil.

The healing ratios of noncapsule beams under dry and water solution healing conditions all improved as the cyclic compression loading time was prolonged. The rationale behind this phenomenon is attributed to the cyclic loading-induced gradual compression of the stones within the asphalt mixture beams, resulting in a marginal diminution of the space between the two parts and a consequent enhancement in the strength rehabilitation rate of the noncapsule beam. It is worth noting that under same loading time, the healing ratio of test beams under dry conditions was higher than that of beams under water solution and the restoration of beam strength was most significantly impacted by the condition of the alkali solution. In summary, water intrusion and salt erosion have a detrimental effect on the self-healing capacity of asphalt concrete in the absence of capsules.

As can be observed from Figure 11, when no external cyclic loads were conducted on the beams, the healing ratio of beams with capsules stayed at a low level (<40%) and improved obviously with the introduction of cyclic compressive loading, which indicates that an external load is necessary to activate the capsules presented in asphalt concrete, which in turn facilitates the release of the encapsulated rejuvenator for improving the healing ratio. Following fixed cyclic loading, the healing capacity of test beams containing capsules exhibited a noticeable improvement in both dry and aqueous solutions when compared to beams lacking capsules. The administration of rejuvenator through capsules

under cyclic loading conditions has the potential to expedite the procedure of crack repair, thereby enhancing the healing efficiency of asphalt concrete.

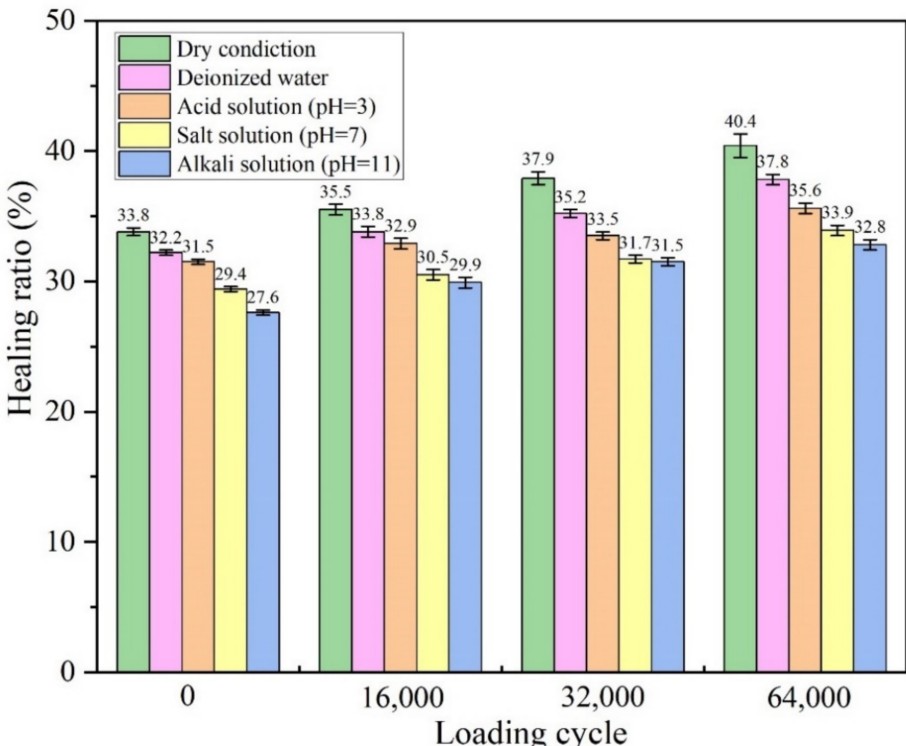

**Figure 10.** The healing level of noncapsule asphalt concrete following different solution treatments.

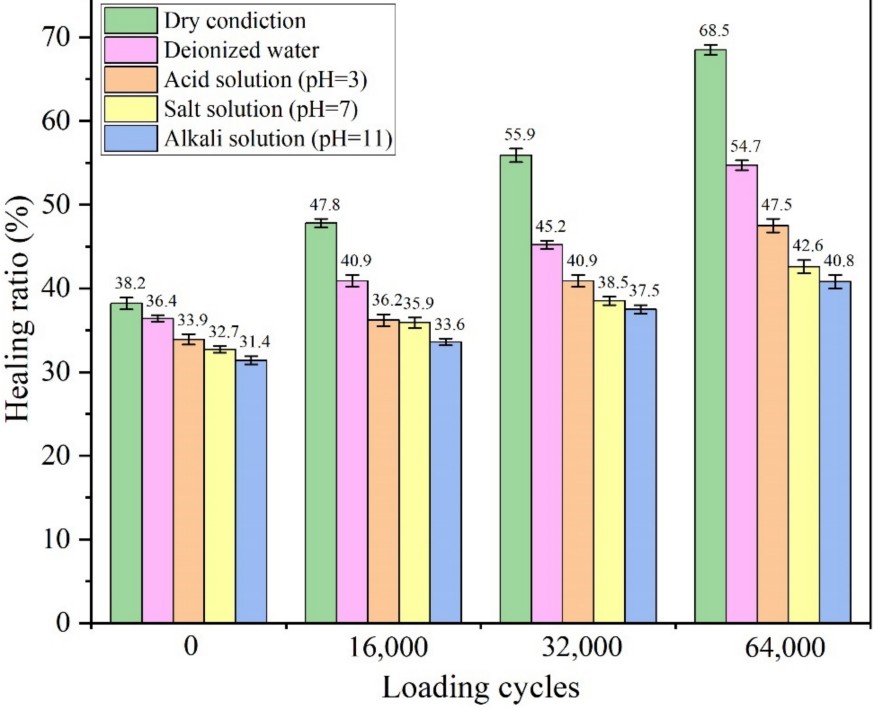

**Figure 11.** The healing level of asphalt concrete with capsules after different solution treatments.

Compared with dry conditions, the healing ratios of test beams under water solution treatment after cyclic loading reduced obviously. Following 64,000 loading cycles, the healing levels of beams containing capsules were evaluated under various conditions, including dry, deionized water, acid, salt, and alkali solution. The obtained results indicate that the healing ratios were 68.5%, 54.7%, 47.5%, 42.6%, and 40.8%, respectively. Additionally, the alkali solution had the most negative impact on the healing ratio of the capsule beams. Although the released rejuvenator aided in the closure of cracks, the water film hindered the flow and diffusion of the rejuvenator in the cracked asphalt area, and of the sodium salt solution into asphalt, forming crystals, which jointly reduced the healing improvement effect of the asphalt rejuvenator.

### 3.2.2. Healing Assessment of Asphalt Concrete after Water Solution Treatment Different Times

The healing rates of asphalt mixture beams without and with capsules after different healing times are shown in Figures 12 and 13, respectively. The loading cycles and healing temperature were set at 64,000 and 20 °C, respectively. Figure 11 illustrates that in the absence of capsules, the healing ratio of the beams exhibited a slight increase as the growth in healing period under dry healing conditions was prolonged. The healing ratios of asphalt concrete without capsules under dry conditions after 48, 72, and 96 h were 40.4%, 41.7%, and 42.5%, respectively. The crack healing ratio is highly dependent on the asphalt flow and diffusion in the crack zone, and more healing time may improve the contact possibility of asphalt in the fractured zone. However, under water solution conditions, the healing ratios of the beams without capsules all showed an obvious reduction tendency as the healing time was prolonged, and the beams under alkali solution conditions exhibited the lowest healing level after the fixed healing time. The healing ratios of asphalt concrete without capsules under deionized water conditions after 48, 72, and 96 h were 37.8%, 36.2%, and 35.0%, respectively, while the healing ratios of asphalt concrete without capsules under alkali solution conditions after 48, 72, and 96 h were 32.4%, 30.4%, and 28.6%, respectively. The presence of water prevents the proximity of asphalt molecules around the crack and slows down the rate of crack closure. Under the action of sodium salt solution, the colloidal equilibrium of asphalt at the crack is destroyed, the stable structure of the asphalt colloid is lost, the interaction force between the components is weakened, and the oil with lighter density undergoes dissolution and diffusion in the erosion process of sodium salt solution [49]. The reason may be that the water intrusion and sodium salt erosion degree gradually aggravated and thus decreased the natural healing ability of the asphalt binder.

In comparison to asphalt mixture beams lacking capsules, the test beams with capsules exhibited a higher healing level after the fixed healing period. For example, the healing ratios of asphalt concrete without and with capsules under dry conditions after the 96 h healing period were 42.5% and 72.4%, respectively. This indicates that the rejuvenator has a notable impact on enhancing the healing capacity of asphalt concrete. The released oil softens the asphalt around the crack area and accelerates the molecular diffusion, which improve the crack repair efficiency of asphalt concrete. The healing rate of the test beams containing capsules exhibited a slight improvement under dry conditions as their healing period was extended. The healing ratios of asphalt concrete without capsules under dry conditions after 48, 72, and 96 h were 68.5%, 70.7%, and 72.4%, respectively, which can be attributed to increased rejuvenator spread across the crack zone. However, when water solutions were introduced into the healing period, the healing rate of the capsule beams decreased as the healing period was prolonged. The healing ratios of asphalt concrete with capsules under deionized water conditions after 48, 72, and 96 h were 54.7%, 52.1%, and 49.6%, respectively. Longer water solution application brought a more serious erosion effect for the asphalt binder in cracked areas and reduced the healing level of the asphalt mixtures. Compared with the deionized water condition, the ion solution had a more severe effect on the healing level of the asphalt concrete. The healing ratios of asphalt concrete with

capsules under alkali solution conditions after 48, 72, and 96 h were 40.8%, 38.1%, and 34.7%, respectively. The presence of moisture and ions prevents the flow of rejuvenating agent around the crack, retarding the contact of asphalt molecules and reducing the rate of crack closure.

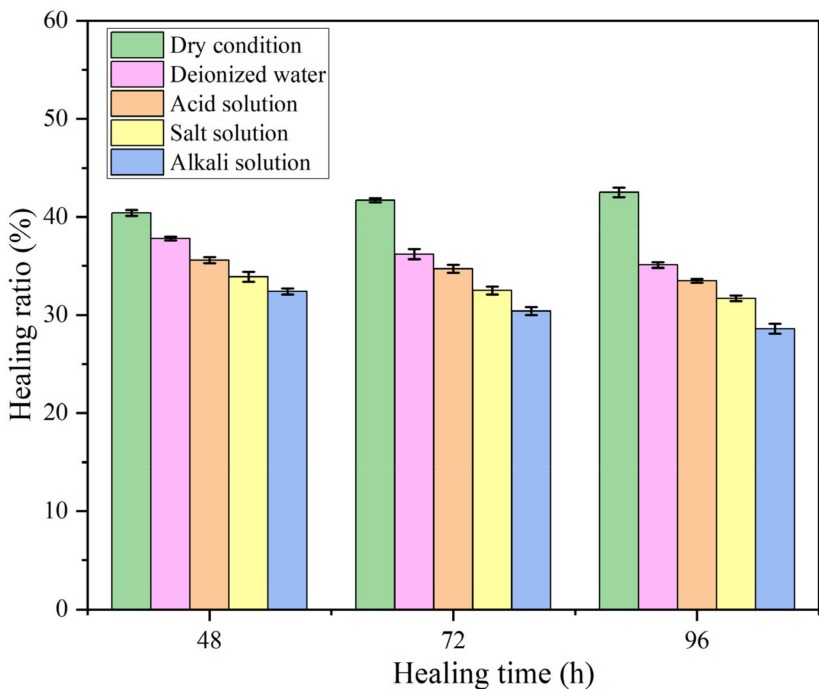

**Figure 12.** The healing rates of noncapsule asphalt concrete after different healing times.

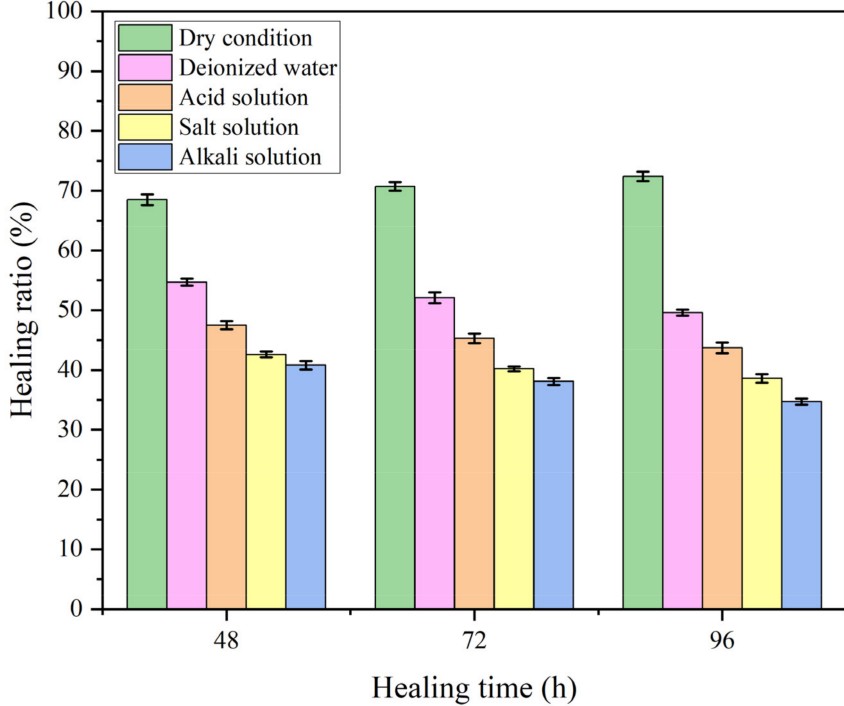

**Figure 13.** The healing rates of asphalt concrete with capsules after different healing times.

*3.3. Rejuvenator Release Ratio of Capsules within Asphalt Concrete under Dry Healing Conditions after Cyclic Loading*

Figure 14 illustrates the oil release percentages for capsules embedded in asphalt concrete subjected to multiple load cycles and subsequent dry conditions. In the absence

of additional load conduction on the beams, it was observed that the oil release ratio amounted to 4.6%. This finding suggests that the capsules had already released a small quantity of rejuvenator during the production of asphalt concrete. When external cyclic loading was applied on the beams, the oil release ratios showed an obvious uptrend and increased with the loading time. This study observed the oil release percentages of capsules embedded in asphalt concrete beams under external loading. The oil release ratios were recorded as 28.5%, 46.2%, and 59.7% after 16,000, 32,000, and 64,000 cycles, respectively. This statement suggests that outside loads are necessary for stimulating the capsules within asphalt concrete, thereby causing the release of the internal rejuvenator. The observed progressive rise in oil release percentage with loads suggests that the capsules possessed long-term release characteristics under cyclic loading, thereby enabling sustainable healing ability for asphalt concrete.

### 3.4. Rejuvenator Content of Asphalt Binder Extracted from Asphalt Concrete under Solution Treatment after Cyclic Loading

After cyclic loading, the test beams were placed under different water solution conditions and the moisture and ion erosion possibly destroyed the asphalt structure and allowed the rejuvenator to flow into the solution environment. Hence, it is necessary to measure the remaining rejuvenator content in the asphalt binder after cyclic loading and subsequent solution treatment. Figure 15 shows the rejuvenator content in asphalt binder after different solution treatments (healing time 96 h and healing temperature 20 °C). After 64,000 cycles of loading, the rejuvenator content in the asphalt binder under dry healing conditions was higher than that of the asphalt binder after water solution treatment, which indicates that the ion solution allowed for more rejuvenator flow release into the water environment. Furthermore, the asphalt binder under alkali solution conditions exhibited the lowest rejuvenator content. The reason may be that the moisture and ion erosion break the structure of the asphalt binder and thus weaken the anchoring effect of the asphalt on the rejuvenator.

### 3.5. Rheological Performance of Extracted Asphalt Binder after Different Solution Treatments

The complex shear modulus (G*) is a parameter that describes the capacity of asphalt to withstand shear deformation, and the phase angle (δ) is a parameter that describes the proportion among the elastic and viscous constituents in bitumen. The viscoelastic behavior of asphalt binder is described using two parameters in a reliant manner. Asphalt binders exhibiting lower G* and higher δ properties are conducive to improved flowability, thereby promoting the repair of cracked asphalt.

Figure 16 presents the rheological index G* and δ of the asphalt binder extracted from asphalt concrete without capsules under different water solution treatments after 64,000 cycles of loading at 20 °C. The G* values of asphalt binder subjected to water solution treatment exhibited a noticeable increase in comparison to those of asphalt binder under dry conditions. The moisture intrusion and sodium salt erosion break the colloidal equilibrium of asphalt and decrease the content of saturated and aromatic fraction and thus increase the proportion of asphaltene, ultimately enhancing the G* of the asphalt binder. Meanwhile, the δ of asphalt binder under water solution treatment were lower than that of asphalt under dry conditions, which indicates that water solution treatment decreases the proportion of viscosity components within asphalt binder and thus increases the content of elastic components. In general, moisture intrusion and sodium salt erosion were found to enhance the G* of asphalt binder while reducing its δ, leading to a reduction in the recuperation capacity of the asphalt binder.

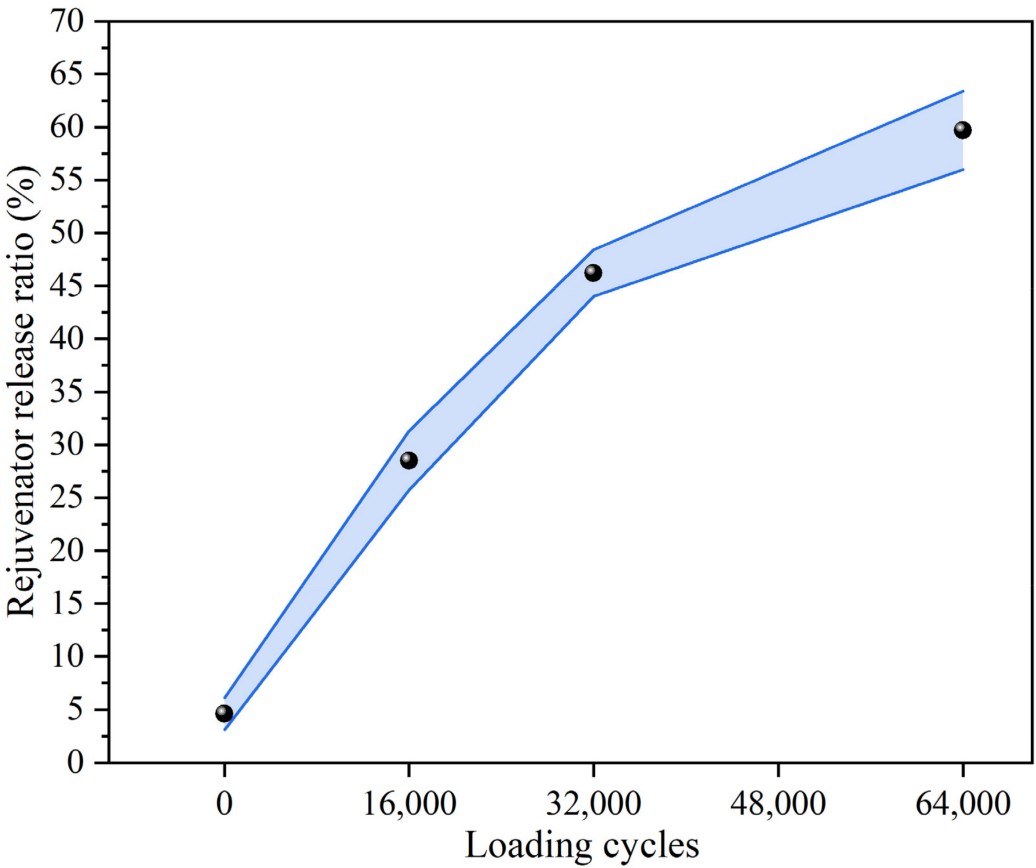

**Figure 14.** Rejuvenator release percentage of capsules within asphalt concrete under dry healing conditions after cyclic loading.

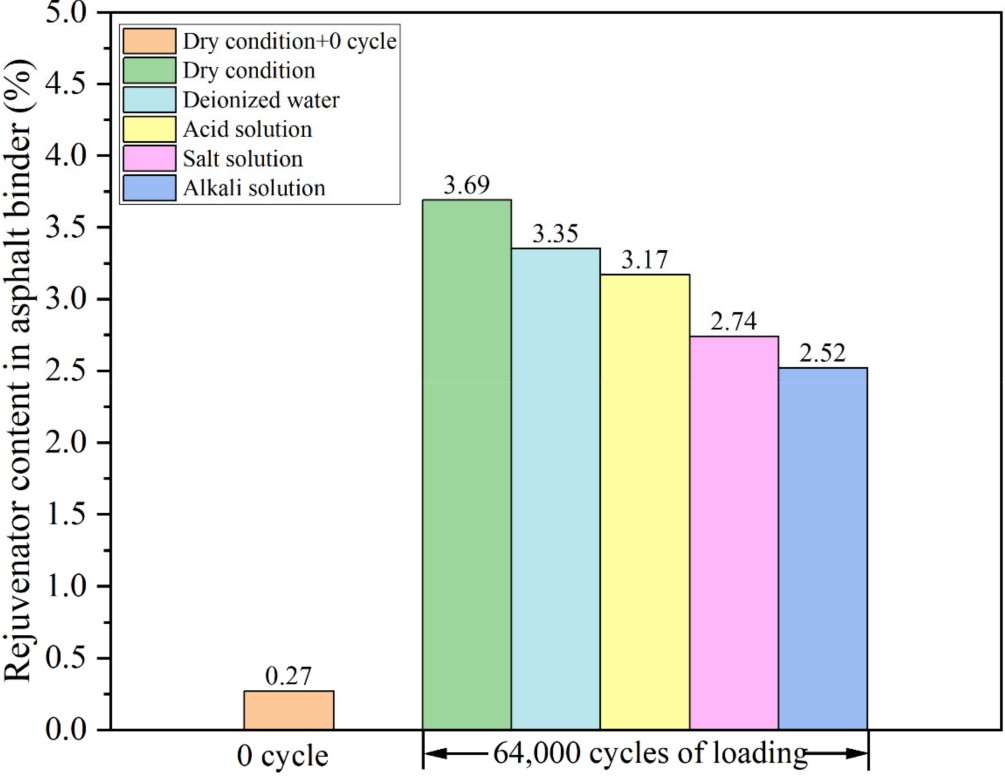

**Figure 15.** The oil content in asphalt binder after different solution treatments.

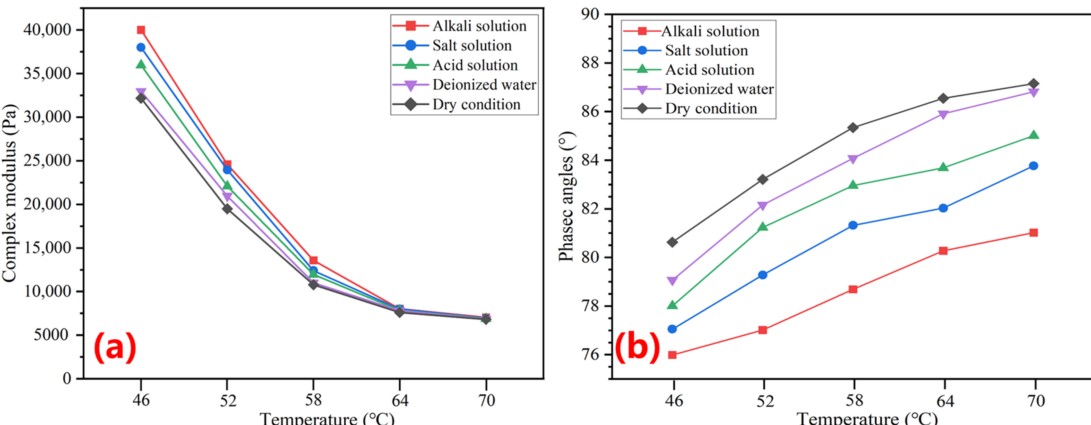

**Figure 16.** The G* (**a**) and δ (**b**) of asphalt binders retrieved from noncapsule asphalt concrete after different solution treatments.

Figure 17 shows the rheological index G* and δ of the asphalt binder extracted from asphalt concrete containing capsules under different water solution treatments after 64,000 cycles of loading at 20 °C. Under dry healing conditions, the asphalt binder extracted from capsule asphalt concrete had lower G* and higher δ than asphalt binder extracted from noncapsule asphalt concrete, which indicates that the released rejuvenator via cyclic loading softened the asphalt binder and improved its flow ability. This study found that the asphalt binder retrieved from capsule asphalt concrete exhibited lower G* and higher δ compared to the asphalt binder extracted from noncapsule asphalt concrete under dry healing conditions. This suggests that the rejuvenator released through cyclic loading had a softening effect on the asphalt binder and enhanced its flow capacity. However, compared with dry healing conditions, the asphalt binder stemming from capsule asphalt concrete under solution treatment still showed higher G* and lower δ, which implies that the moisture and salt erosion decreased the flow ability enhancement effect of the asphalt rejuvenator.

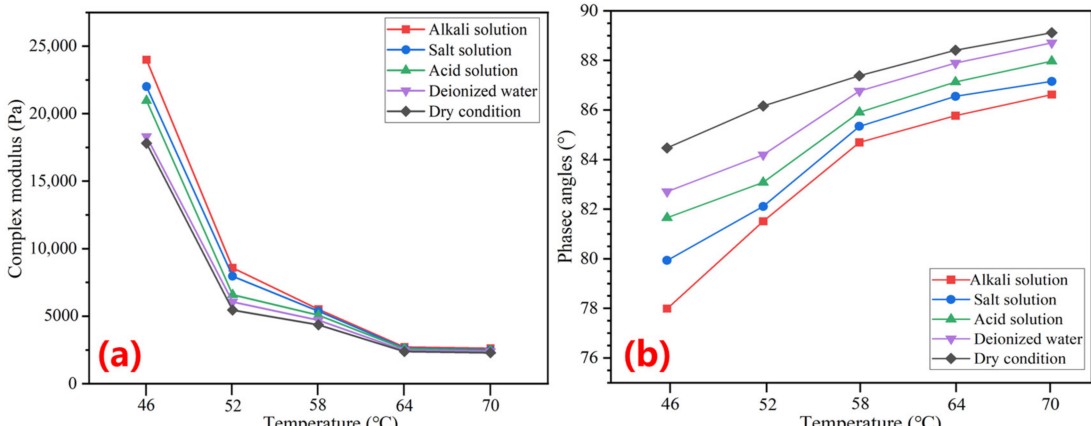

**Figure 17.** The G* (**a**) and δ (**b**) of asphalt binders retrieved from asphalt concrete with capsules after different solution treatments.

## 4. Conclusions

Polymer capsules that encapsulated rejuvenator were synthesized. The capsules underwent a battery of assessment tests to investigate their morphological framework, mechanical strength, thermal stability, and rejuvenator loading content. This study assessed the healing rates of asphalt concrete containing capsules under various solution healing conditions. This study involved the determination of the oil discharge percentage of capsules under dry conditions, and the remaining rejuvenator contents in asphalt binder

after solution treatment were determined. Furthermore, the rheological performance of extracted asphalt binder was investigated. The statements lead to the following conclusions:

(1) The polymer capsules exhibit an internal multichamber structure and demonstrate favorable mechanical and thermal durability, satisfying the criteria for use in laboratory asphalt concrete production.

(2) The capsules with multiple chambers exhibited a gradual rejuvenator release pattern, thereby imparting long-term healing potential to asphalt concrete. The solution healing condition decreased the remaining rejuvenator in the extracted asphalt binder after being subjected to cyclic loading.

(3) The utilization of capsules in asphalt concrete resulted in higher healing ratios compared to noncapsule asphalt concrete, both under dry and solution healing conditions, due to the released rejuvenator via cyclic loading. Meanwhile, the healing efficacy of asphalt concrete, both with and without capsules, was found to be diminished by the moisture intrusion and ion erosion resulting from the solution treatment. The alkali solution has the most adverse effect on the enhancement in healing efficacy for asphalt concrete.

(4) The healing efficacy of asphalt concrete containing capsules in a dry curing environment increased marginally as the healing duration was prolonged, while the asphalt concrete with capsules exhibited lower healing efficacy after a longer healing period due to the severe erosion effect in the crack zone.

(5) The simultaneous impact of moisture intrusion and ion erosion was observed to enhance the complex modulus of asphalt binder while reducing its phase angle, thereby diminishing the healing capacity of the asphalt binder.

The asphalt concrete with polymer capsules subjected to dry and water healing conditions achieved a good healing ratio due to the released rejuvenator; however, with the intrusion of ion solution, the healing level reduced sharply due to the joint effect of moisture and ion erosion. We recommend that the polymer capsules are incorporated into asphalt concrete in low-rain regions. The present study examined the curative characteristics of asphalt concrete in the absence and presence of polymer capsules under diverse solution healing conditions. We recommend that subsequent research endeavors focus on the impact of the solution on the structural transformation through atomic force microscopy and the process of crack closure through molecular dynamics simulation.

This paper explored the healing properties of asphalt concrete with capsules under static erosion under different solution conditions. However, in the actual service process of pavement, rain erosion is a dynamic process, and under the influence of vehicle loading, the healing performance of asphalt concrete is more difficult to predict. This work does not consider the effect of dynamic erosion on the healing properties of asphalt concrete with self-healing capsules in water environments. Hence, it is recommended that future research be concentrated on the healing level of asphalt concrete with capsules under dynamic water erosion conditions using a modified Hamburg wheel tracking test.

**Author Contributions:** Z.L.: Investigation, Conceptualization, Methodology, Writing—Original Draft. H.W.: Investigation, Project Administration. P.W.: Supervision, Writing—Review and Editing. Q.L.: Writing—Review and Editing. S.X.: Formal analysis. J.J.: Writing—Review and Editing. L.F.: Writing—Review and Editing. L.T.: Writing—Review and Editing. All authors have read and agreed to the published version of the manuscript.

**Funding:** This research was funded by the National Natural Science Foundation of China (No. 52108416), the National Natural Science Foundation of China (No. 52378461),the National Natural Science Foundation of China (No. 52308464), Guangdong Basic and Applied Basic Research Foundation (2023A1515011448) and Hubei Science and Technology Innovation Talent and Service Project (No. 2022EHB006).

**Institutional Review Board Statement:** Not applicable.

**Informed Consent Statement:** Not applicable.

**Data Availability Statement:** Data is contained within the article.

**Conflicts of Interest:** The authors declare no conflict of interest.

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
