# Peer review of "Healing Evaluation of Asphalt Mixtures with Polymer Capsules Containing Rejuvenator under Different Water Solutions"

_sustainability, doi:10.3390/su152115258_

Round 1

Reviewer 1 Report

Article "Healing evaluation of asphalt mixtures with polymer capsules containing rejuvenator under different water solutions" is devoted to the performance of asphalt binders. The article is written accurately and conforms to the scientific style of writing. The following questions and comments were found during the study of this article:

- in the introduction there is no mention of which modifications lead to improved properties of bitumen, e.g. addition of polymers, biomaterials, nanoparticles, etc. (10.1016/j.carbpol.2023.120896, 10.1533/9780857093721.1.72). In addition, examples of improving the properties of bitumen with light components, e.g. from bio-oil (10.1016/j.conbuildmat.2022.127946), should be given in line 75-77 sentence.

- line 103 - typo "Nonetheless, However"

- for the original bitumen it is important to specify viscosities at 25°C and 180°C

- the meaning of the sentence "Since temperature has a impact on the mechanical strength." is unclear. It might be worth rewriting it.

- if possible, it would be interesting to consider the dependence of the rutting factor (G*/sinδ = G*2/G") on temperature for the all modified bitumens.

Author Response

Article "Healing evaluation of asphalt mixtures with polymer capsules containing rejuvenator under different water solutions" is devoted to the performance of asphalt binders. The article is written accurately and conforms to the scientific style of writing. The following questions and comments were found during the study of this article:

1. in the introduction there is no mention of which modifications lead to improved properties of bitumen, e.g. addition of polymers, biomaterials, nanoparticles, etc. (10.1016/j.carbpol.2023.120896, 10.1533/9780857093721.1.72). In addition, examples of improving the properties of bitumen with light components, e.g. from bio-oil (10.1016/j.conbuildmat.2022.127946), should be given in line 75-77 sentence.

Response: Thank you for the comment. In the introduction, we have added the description of modified asphalt in the revised manuscript. “To improve the healing ability of asphalt, some researcher tried to add polymer, bio-materials, nanoparticle into asphalt [17, 18].” In the revised manuscript, we have added the reference into it.“This technology employs healing agent that comprises abundant light asphalt components, which facilitate the restoration of the unique natural healing features of the aged bitumen binder[31].”We have added the three references rationally in the revised manuscript.

2. line 103 - typo "Nonetheless, However".

Response: Thank you for the kind remind. We have revised it.

3. for the original bitumen it is important to specify viscosities at 25°C and 180°C

Response: Thank you for the comment. In this work, the used asphalt is base 60/80 bitumen. In general, we give the dynamic viscosity of asphalt at 60°C. we have added the viscosity of asphalt in the revised manuscript. “This study reports the values of density, viscosity, penetration, and softening point for virgin asphalt (#70) as 1.034g/cm3, 231.3 Pa·s(60°C), 68(0.1mm, 25°C), and 48.4°C respectively.”

4. the meaning of the sentence "Since temperature has a impact on the mechanical strength." is unclear. It might be worth rewriting it.

Response: Kind remind. We used the find function inside “word” and did not find the sentence“Since temperature has a impact on the mechanical strength”.

5. if possible, it would be interesting to consider the dependence of the rutting factor (G*/sinδ = G*2/G") on temperature for the all modified bitumens.

Response: Thank you for the valuable comment. In this paper, the complex modulus and phase angle were used to reflect the rheological property of asphalt binder extracted from asphalt concrete with capsules.

Reviewer 2 Report

The healing properties of bituminous concrete containing polymer capsules were investigated in this research under various solution healing conditions after cyclic loading. This study is innovative, experimentally adequate and the paper is well-written. However, the following questions should be clarified and some points should be taken into account.

- The objectives should be written clearly.

-line 103, use either Nonetheless, or However>

- please add the properties of bitumen and sunflower in tables.

- line 135, please introduce the abbreviation of "SA"

line 138, use "rpm" instead of revolutions per minute

-How were the rates determined in line 136, 140, 141, 142 of the capsule Preparation ?

- line 161 , please use asphalt concrete instead of bitumen concrete

Author Response

The healing properties of bituminous concrete containing polymer capsules were investigated in this research under various solution healing conditions after cyclic loading. This study is innovative, experimentally adequate and the paper is well-written. However, the following questions should be clarified and some points should be taken into account.

1. The objectives should be written clearly.

Response: Thank you for the valuable comment. We have highlighted the objective of this paper in the revised manuscript. “This study aimed at exploring the healing levels of asphalt concrete containing polymer capsules under various solution healing condition following cyclic loads”.

2. line 103, use either nonetheless, or however>

Response: Thank you for the kind remind. We have revised it.

3. please add the properties of bitumen and sunflower in tables.

Response: Thank you for the kind remind. We have added the basic information of base 60/80 asphalt and sunflower oil in the revised manuscript. “This study reports the values of density, viscosity, penetration, and softening point for virgin asphalt (#70) as 1.034g/cm3, 231.3 Pa·s (60°C), 68 (0.1mm, 25°C), and 48.4°C respectively. The basic information of asphalt rejuvenator sunflower oil was presented in the table 2.”

Table 2. The Basic properties of sunflower oil used in this work.

Item

Value

Appearance

Light yellow liquid

Main component

Fatty acid

Density (15°C)

0.935 g/cm3

Viscosity (60°C)

0.285 Pa⋅s

Flash point

230°C

4. line 135, please introduce the abbreviation of "SA"

Response: Thank you for the kind remind. The SA represented “sodium alginate”. We have added the full name in the revised manuscript. “A 2.25wt% even sodium alginate (SA) solution of was prepared by adding sodium alginate powder into water”.

5. line 138, use "rpm" instead of revolutions per minute

Response: Thank you for the kind remind. We have revised the unit of shear speed. “Followed by subjecting the mixture to shear at 5000 rpm for a duration of 15 minutes”.

6. How were the rates determined in line 136, 140, 141, 142 of the capsule preparation?

Response: Thank you for the valuable comment. The appropriate fabrication parameter range of calcium alginate capsule has been determined in previous research. Hence, in this work, we used specific capsules preparation parameter.

7. line 161, please use asphalt concrete instead of bitumen concrete

Response: Thank you for the kind remind. We have replaced asphalt concrete instead of bitumen concrete in the whole paper.

Reviewer 3 Report

This manuscript explored the healing attributes of bitumen concrete containing polymer capsules under various solution healing conditions following cyclic loads. Some issues should be addressed.

1.        What is the novelty of this work? What progress rather than previous studies can be highlighted? I think it is not obvious.

2.        The specific information on SEM images in Fig. 5 is too small.

3.        What's the horizontal axis in Fig. 7?

4.        Some essential background and typical investigations should be added in the introduction to provide a complete literature review. Such as: Optimal water-cement ratio of cement-stabilized soil. Construction and Building Materials, 320, 126211. Meso-mechanical investigations on the overall elastic properties of multi-phase construction materials using finite element method. Construction and Building Materials, 228, 116727.

5.        The language used in this manuscript also needs to be thoroughly revised (such as repeated sentences, black, …).

6.        The legend in Fig. 8 should be provided.

7.        What's the role of Figs. 11-12? To give readers a clear understanding of this work, we suggest the author reorganize the associated texts by providing a deeper, more thoughtful analysis.

The language used in this manuscript also needs to be thoroughly revised (such as repeated sentences, black, …). 

Author Response

This manuscript explored the healing attributes of bitumen concrete containing polymer capsules under various solution healing conditions following cyclic loads. Some issues should be addressed.

1. What is the novelty of this work? What progress rather than previous studies can be highlighted? I think it is not obvious.

Response: Thank you for the valuable comment. The current research about healing property of asphalt concrete with calcium alginate capsule capsules are conducted in dry healing environment. Due to the asphalt pavement in the actual service process by the influence of the external service environment, such as rainy areas, acidic areas, coastal areas, and saline and alkaline areas, etc., the adverse environment will accelerate the aging of asphalt, so that the asphalt concrete earlier disease, such as cracking, etc. Hence it is vital to study the effect of different water environments on the self-repair performance of asphalt concrete containing calcium alginate capsule. We have highlighted the novelty of this work in the revised manuscript.

2. The specific information on SEM images in Fig. 5 is too small.

Response: Thank you for the valuable comment. We have added SEM picture of capsule in the revised manuscript. Meanwhile, we have added specific information in the SEM image.

3. What's the horizontal axis in Fig. 7?

Response: Thank you for the kind remind. We have added the horizontal axis in Fig.7.

4. Some essential background and typical investigations should be added in the introduction to provide a complete literature review. Such as: Optimal water-cement ratio of cement-stabilized soil. Construction and Building Materials, 320, 126211. Meso-mechanical investigations on the overall elastic properties of multi-phase construction materials using finite element method. Construction and Building Materials, 228, 116727.

Response: Kind comment. We have downloaded the two documents you recommended and read the literature carefully. Your paper is of excellent quality and highly readable. However, we found that the research in the literature you recommended is irrelevant to the research topic of this paper. We think twice and decided to give up the quote of the two literatures.

5. The language used in this manuscript also needs to be thoroughly revised (such as repeated sentences, black, …).

Response: Thank you for the kind remind. We have checked the whole paper and polished the language carefully.

6. The legend in Fig. 8 should be provided.

Response: Thank you for the kind remind. We have added the legend in Fig.8 in the revised manuscript.

7. What's the role of Figs. 11-12? To give readers a clear understanding of this work, we suggest the author reorganize the associated texts by providing a deeper, more thoughtful analysis.

Response: Thank you for the constructive comment. The healing time have impact on the healing ratio of asphalt concrete. Fig. 11 and 12 presented the healing level of asphalt concrete without and with capsules in different solution condition after various curing time. We have graphed the data to present the most intuitive trend in healing rates. Meanwhile, we have added deep analysis of two figures.

Reviewer 4 Report

 This paper has evaluated the healing of asphalt mixtures with polymer capsules containing rejuvenator under different water solutions. The topic considered under this paper is interesting. Authors have a made a great contribution and have shown that they have understand the topic. However, there are some issues they should address for consideration to publication.

·           Authors should clearly present the research gaps, motivations, and contributions in the introduction.

·         Authors should indicate why they consider to work in this topic?

·         I suggested authors to draw a flowchart that will show the whole process of methodology used.

·         In the conclusion, authors should indicate the limitations of their study and give more recommendations for further studies.

Minor editing of English is required

Author Response

This paper has evaluated the healing of asphalt mixtures with polymer capsules containing rejuvenator under different water solutions. The topic considered under this paper is interesting. Authors have a made a great contribution and have shown that they have understand the topic. However, there are some issues they should address for consideration to publication.

1. Authors should clearly present the research gaps, motivations, and contributions in the introduction.

Response: Thank you for the constructive comment. We have highlighted the research gaps, motivations, and contribution in introduction section in the revised manuscript.

2. Authors should indicate why they consider to work in this topic?

Response: Thank you for the profound comment. The ongoing researches about healing level of asphalt mixtures with calcium alginate capsules are conducted under con-trolled non-aqueous healing conditions. Due to the asphalt pavement in the actual service process by the influence of the external service environment, such as rainy areas, acidic areas, coastal areas, and saline and alkaline areas, etc., the adverse environment will accelerate the aging of asphalt, so that the asphalt concrete earlier disease, such as cracking, etc. Hence it is vital to study the effect of different water environments on the self-repair performance of asphalt concrete containing calcium alginate capsule.

3. I suggested authors to draw a flowchart that will show the whole process of methodology used.

Response: Thank you for the valuable comment. Followed your kind advice, we have added research flowchart in the revised manuscript.

4. In the conclusion, authors should indicate the limitations of their study and give more recommendations for further studies.

Response: Thank you for the profound comment. We have added the limitation of this study and recommendation for further studies in the revised manuscript.

“The asphalt concrete with polymer capsules subjected to dry and water healing condition obtained good healing ratio due to the released rejuvenator, however, with the intrusion of ion solution, the healing level reduced sharply due to the joint effect of moisture and ion erosion. It recommends that the polymer capsules are incorporated into asphalt concrete under low-rain regions. The present study examined the curative characteristics of asphalt concrete in the absence and presence of polymer capsules under diverse solution healing conditions. It recommends that subsequent research endeavors can focus on the impact of the solution on the structural transformation through atomic force microscopy and the process of crack closure through molecular dynamics simulation.”

“This paper explored the healing properties of asphalt concrete with capsules un-der static erosion in different solution condition. However, in the actual service process of pavement, the rain erosion is a dynamic process, and under the action of vehicle loading, the healing performance of asphalt concrete is more difficult to predict. This work does not consider the effect of dynamic erosion on the healing property of asphalt concrete with self-healing capsules under water environment. Hence, it is recommended that the future research can be concentrated on the healing level of asphalt concrete with capsules under dynamic water erosion condition through modified Hamburg Wheel-Tracking test.”

Round 2

Reviewer 1 Report

The authors made the necessary edits, therefore the article can be published without further changes

Reviewer 4 Report

Authors have addressed all my comments. And the paper has been significantly improved. Based on that, I recommend it to be published.